# The Northern Adriatic Forecasting System for Circulation and Biogeochemistry: Implementation and Preliminary Results

Isabella Scroccaro [1,2], Marco Zavatarelli [1,*], Tomas Lovato [3], Piero Lanucara [4] and Andrea Valentini [5]

1 Dipartimento di Fisica e Astronomia, Alma Mater Studiorum Università di Bologna, Viale Berti Pichat 8, 40127 Bologna, Italy
2 Regional Agency for the Protection of the Environment of Friuli Venezia Giulia, Via Delle Acque 28, 33170 Pordenone, Italy
3 Fondazione Centro Euro-Mediterraneo sui Cambiamenti Climatici, CMCC, Viale Berti Pichat 6/2, 40127 Bologna, Italy
4 SuperComputing Applications and Innovation Department, CINECA, Via Dei Tizii 6, 00185 Roma, Italy
5 Arpae Emilia-Romagna, Hydro-Meteo-Climate Service (Arpae-SIMC), Viale Silvani 6, 40122 Bologna, Italy
* Correspondence: marco.zavatarelli@unibo.it

**Abstract:** This paper described the implementation of a forecasting system of the coupled physical and biogeochemical state of the northern Adriatic Sea and discussed the preliminary results. The forecasting system is composed of two components: the NEMO general circulation model and the BFM biogeochemical model. The BFM component includes an explicit benthic pelagic coupling providing fluxes at the sediment–water interface and the dynamic of the major benthic state variables. The system is forced by atmospheric forcing from a limited-area model and by available land-based (river runoff and nutrient load) data. The preliminary results were validated against available remote and in situ observations. The validation effort indicated a good performance of the system in defining the basin scale characteristics, while locally the forecasting model performance seemed mostly impaired by the uncertainties in the definition of the land-based forcing.

**Keywords:** forecasting system; environmental dynamics; northern Adriatic Sea; benthic pelagic coupling





## 1. Introduction

Coastal interdisciplinary oceanography environmental dynamics are governed by a suite of external physical and biogeochemical drivers, whose reciprocal interactions originate complex patterns of ecosystem functioning [1–3].

The northern Adriatic Sea (hereafter NAD), see Figure 1, is the northernmost region of the Mediterranean Sea and is a clear example of these kinds of interactions. It is a truly coastal (epicontinental) basin with an average depth of about 35 m. Differently from the rest of the Mediterranean Sea, it is affected by a significant river runoff [4], which qualifies most of the basin as a region of freshwater influence (ROFI) [5]. The atmospheric (wind stress and heat/water fluxes) and land-based (river runoff) components drive an overall cyclonic circulation marked by a pronounced seasonality and a strong interannual variability [6,7]. Most notably, the strong heat losses occurring in winter may give rise (despite the freshening due to the strong river runoff) to the formation of shelf dense water [8–10], which moves along the bottom toward the southernmost region of the Adriatic Basin. The environmental dynamics of the basin are clearly influenced by the land-based nutrients load associated with the river runoff [11]. The river runoff load can determine meso- to eutrophic conditions, enhanced in the past by the intense anthropogenic pressure. This pressure gave rise to serious environmental problems such as the development of extended (in space and time) anoxic events along the western NAD coast [12]. However, the severity of the eutrophic events appears to have been recently mitigated [13,14].

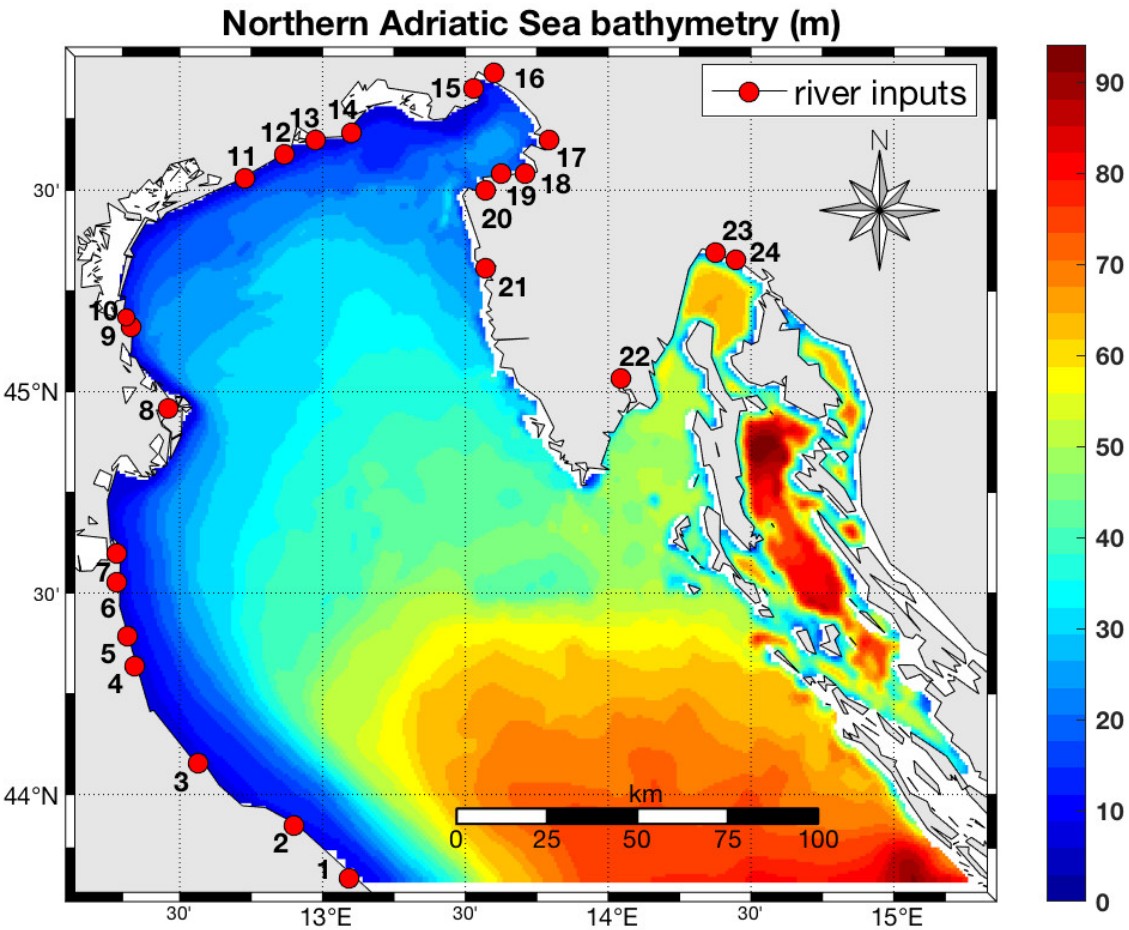

**Figure 1.** The northern Adriatic Sea model domain and bathymetry. The red dots indicate the rivers discharging into the basin. 1: Metauro, 2: Foglia, 3: Marecchia, 4: Savio, 5: Fiumi Uniti, 6: Lamone, 7: Reno, 8: Po, 9: Adige, 10: Bacchiglione, 11: Piave, 12: Livenza, 13: Canale Lovi, 14: Tagliamento, 15: Isonzo, 16: Timavo, 17: Rizana, 18: Drnica, 19: Dragonia, 20: Badasevica, 21: Mirna, 22: Rasa, 23: Rjecina, 24: Ivan.

This paper reported the initial development and the first results obtained from the implementation of an operational forecasting system of both the physical and biogeochemical state of the NAD (hereafter NAD-FC). It is believed that, given the physical and environmental dynamics sketched above, as well as the mentioned anthropogenic pressure, a truly interdisciplinary oceanographic forecasting system specifically designed for the NAD might represent a valuable tool to support scientific and policy-related issues. Given the NAD characteristics of a shallow basin, the modeling system—developed to routinely provide analysis and forecast of the environmentally relevant properties of the basin—extends beyond the pelagic domain by fully including the dynamics of the benthic system. This is achieved by implementing an explicit "benthic pelagic coupling" [15], accounting for the sediment biogeochemical cycling fully connected with the homologous cycling processes occurring in the water column. This is deemed to be crucial when dealing with a shallow basin such as the NAD, where the temporal and spatial variability of the benthic–pelagic interactions plays a relevant role in defining the overall marine ecosystem functioning.

The paper is structured in order to provide a description of the numerical modeling system components constituting the NAD-FC, as well as the coupling among them. The implementation over the NAD domain, the forcing fields, and the relative boundary conditions used to provide the alternate production of analysis and forecast (the operational chain) are then offered. A preliminary assessment of the model skill is provided through the comparison against different observational datasets and with respect to variables that are of

interest to potential users: seawater temperature and salinity, as well as dissolved oxygen and chlorophyll. In particular, observations account for data from satellites, autonomous buoys, and vertical profiles.

The model configuration and observation data sets used are presented in Section 2. Section 3 presents the first model assessment, followed by discussion and conclusions in Section 4.

## 2. Materials and Methods

### 2.1. The Components of the NAD-FC

The NAD-FC is composed of the explicit online coupling of two ocean numerical models, namely a general circulation model and a biogeochemical model.

The general circulation model component is based on the NEMO model version 3.6 [16]. NEMO is a numerical ocean model adopting the primitive Navier-Stokes equations and a non-linear equation of state coupling active tracers (temperature and salinity) to the fluid velocity. The model is built on the Boussinesq, hydrostatic, incompressibility, and turbulence closure hypotheses and adopts the spherical earth and the thin-shell approximations.

The space discretization is based on an Arakawa-C grid type. The numerical solutions of the governing equations are based on the source splitting method [17] operating on a time marching step of 120 s. Temperature, salinity, and the biogeochemical tracers were advected with a mixed upstream/MUSCL numerical scheme: a monotonic upwind scheme for conservation laws [18], originally implemented by Estubier and Levy [19] and modified by Oddo et al. [20]. Horizontal diffusion of momentum and tracers were modeled with a Bi-Laplacian formulation with enhanced vertical viscosity and diffusivity. The vertical turbulence was modeled with the generic length scale model [21,22] using k-$\varepsilon$ closure and Canuto stability functions. Closed lateral boundaries were handled by a no-slip condition, while a nonlinear formulation was used for the bottom friction.

The biogeochemical model component was the Biogeochemical Fluxes Model, BFM [23,24], an open-source community biogeochemical model. BFM is a generic biomass and functional group-based marine ecosystem model. It represents the system in Eulerian coordinates by a suite of chemical and biological processes simulating the pelagic (water column) and benthic (sediment) dynamics of the marine ecosystem. A scheme of the relationships among the model state variables is given in Figure 2. In both the pelagic and benthic domains the elemental components (carbon, nitrogen phosphorus, silicon, and oxygen) exchanged by the pelagic/benthic state variables are organized in Chemical Functional Families (CFF: inorganic, living, and non-living organic). The living organic CFF also identify the Living Functional Groups (LFG). The biogeochemical cycling is solved independently across nutrients, carbon dioxide, oxygen, diatoms, nano-, pico- and large-phytoplankton, micro-, and meso-zooplankton, heterotrophic nano-flagellates, aerobic/anaerobic bacteria (pelagic and benthic), filter feeders, infaunal predators, detritus feeders, Meio- and Macro-benthos, and particulate and dissolved organic matter (see Figure 2a,b for the pelagic and benthic components, respectively). A full description of the equations governing the pelagic biogeochemical dynamics can be found in Vichi et al. [24]. It is stressed that the pelagic domain model structure fully resolves the so-called "herbivorous" [25,26] and "microbial" web pathways [27,28]. The two configuration pathways are to be considered as the extremes of a trophic continuum [29]. In BFM the prevailing path(s) of matter flow can shift between these two extremes and is (are) modulated by the space–time-dependent variability of the environmental characteristics [30].

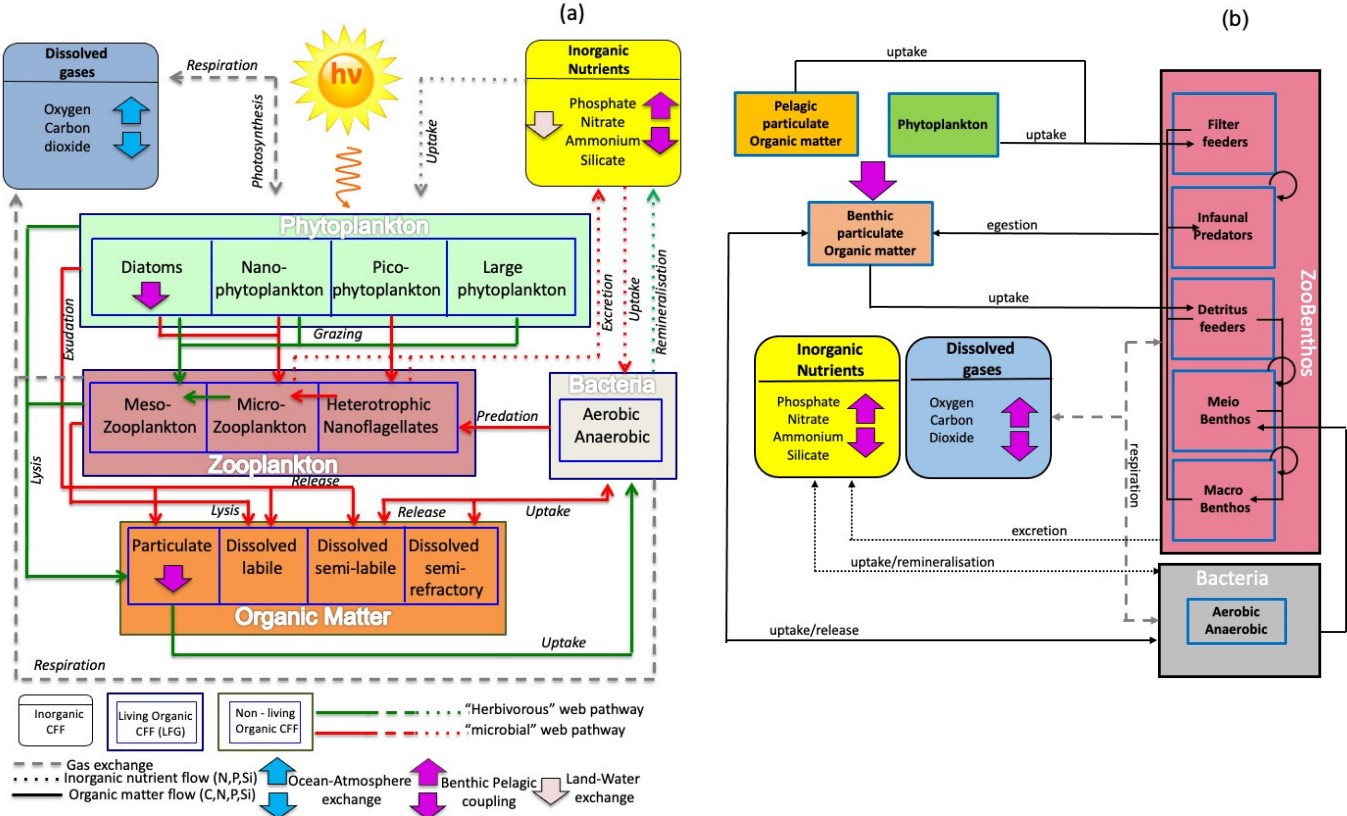

**Figure 2.** The BFM model's state variables and structure. LFG's exchange carbon, nitrogen, phosphorus, silicon, and oxygen trough flow of CFF's. (**a**) The pelagic component. Green arrows represent the herbivorous web and the red arrows the microbial web. (**b**) The benthic component.

The BFM benthic sub-model provides a detailed representation of the benthic biogeochemical cycling along with a direct coupling between the pelagic and biogeochemical cycling processes. The benthic model component (Figure 2b) is based on the model effort of Ebenhöh et al. [31] and Ruardij and Raaphorst [32]. The sediment's vertical structure resolves two dynamical layers (oxic and anoxic) where different processes take place. The organic matter has an implicit vertical distribution and the sediment oxygen dynamics are resolved including the dynamical shifting of the oxic layer.

The benthic–pelagic coupling is defined by particulate organic matter and phytoplankton sedimentary flux, diffusive inorganic carbon, and nutrients and oxygen fluxes. At the water sediment interface, a specific sinking (burial) velocity is defined. All these processes are described in Mussap and Zavatarelli [33], whose one-dimensional implementation was here extended to the three-dimensional case.

## 2.2. The Model Coupling

NEMO and BFM are synchronously coupled online and share the same time-marching step. The temporal integration of the BFM state variables is based on the source-splitting technique [34]. The biogeochemical rate of change is directly inserted into the transport tracer integration. For each biogeochemical state variable, the horizontal/vertical rate of change due to the advection and the horizontal diffusion are computed first and the biogeochemical rate of change is added prior to an intermediate time integration. The result of this operation is then passed to an implicit Euler backward solver for the vertical diffusion. The advection and diffusion numerical schemes applied to the BFM state variables are the same as those used by NEMO for the physical tracers.

### 2.3. Model Implementation

The model domain covers the northern Adriatic Sea, spanning from about 12° W to 16° E and 43.5° to 46° N (Figure 1), whose general morphological, hydrographical, and environmental characteristics were summarized in Section 1. The sea area covered by the numerical grid was about $3 \times 10^4$ km$^2$ and the maximum depth in the model domain was about 95 m. The horizontal grid resolution was approximately 800–900 m and the only open boundary was located at the southern limit of the domain. The vertical resolution was ensured by 47 z-layers in the vertical direction, defining a constant layer thickness of 2 m. The model bathymetry was obtained through a bilinear interpolation of the GEBCO 2021 gridded bathymetry data [35].

#### 2.3.1. Surface, Land-Based, and Open Boundary Conditions

The modeling system interactively computes atmosphere–ocean surface fluxes of momentum, mass, and heat using the bulk formulae described in Castellari et al. [36]. The hourly atmospheric data (2 days of analysis and forecast) used for the surface fluxes computation (mean sea level pressure, total cloud cover, 10 m wind velocity, 2 m air and dew point temperature, total precipitation-averaged field) originated from the COSMO operational model [37], operated by the meteorological forecasting service of the Agency for Prevention, Environment and Energy of the Emilia–Romagna Region (Arpae). The COSMO model has a spatial horizontal resolution of 2.2 km and 65 vertical σ-layers. The oxygen and carbon dioxide exchanges occurring at the ocean–atmosphere interface were parameterized according to Wanninkhof [38,39].

The model implementation considers the runoff contribution from 24 rivers (see Figure 1). The Po river, with a climatological runoff of about 1500 m$^3$/s, roughly accounts for 30% of the total Adriatic Sea fresh water runoff and represents the largest river discharging into the NAD. The Po river runoff is monitored daily by the Arpae river flow-monitoring network. The selected station is located at the closing section of the Po drainage basin, namely in Pontelagoscuro, at approximately 50 km from the delta shores. For the other Adriatic rivers, the discharge is climatologically defined at a monthly resolution, using the data from Ludwig et al. [40]. Salinity of riverine water was set to 15 psu, similarly to previous modeling applications for the entire Adriatic Sea, see Zavatarelli and Pinardi [10] and Oddo et al. [41]. The river runoff is associated with a nutrient river load for nitrate, phosphate, and silicate, which was resolved by using a monthly climatology based on Ludwig et al. [40].

At the southern open boundary, the model's physical component is one-way off-line nested with the Mediterranean Sea forecast system data [42], distributed by the Copernicus Marine Environment Monitoring Service (CMEMS). Data for sea surface elevation, depth-averaged velocity, and vertical profiles of temperature and salinity were interpolated and specified onto the boundary to provide the open boundary conditions, which were computed according to Oddo and Pinardi [43], using Orlansky [44] and Flather's [45] schemes. The same open boundary scheme adopted for temperature and salinity was applied to the biogeochemical tracers.

Initial and open boundary data for the biogeochemical tracers were obtained from climatology of the environmental dynamics of the whole Adriatic basin. This climatology was computed from a previous 30-year modeling hindcast effort, carried out with a coupled modeling system composed of the Princeton Ocean Model and the BFM [10–46].

### 2.4. The Operational Chain

The model began its operational activity on 15 October 2021. This paper is based on the forecasting system results spanning the period November 2021–April 2022. At the very beginning of the operational activity, the first forecasting cycle was initialized from a restart originated by a long (10-year) hindcast exercise.

Operational activities were based as usual on alternating analysis and forecast runs as depicted in Figure 3. Every day ("D" in Figure 3), a forecast run was launched from a

restart of the previous analysis run. The forecast run was forced with the 2 day atmospheric and open boundary forecast data described above. The last known Po river runoff daily data persisted throughout the entire forecast run.

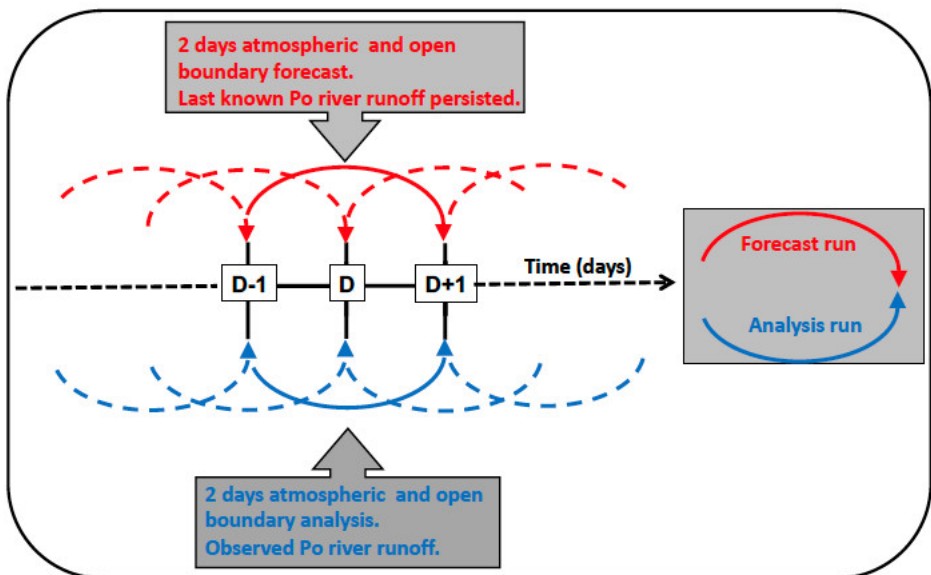

**Figure 3.** The structure of the operational chain.

The analysis run followed the same procedure of the forecast run, but utilized the atmospheric and open boundary analysis data. In this case, the observed daily runoff of the Po river was applied.

Forecast and analysis runs were executed in parallel mode using 176 processors and 4 nodes on the G100 supercomputer of the CINECA interuniversity consortium. The operational chain shown in Figure 3 was run in semi-automatic mode, as the whole procedure was started manually and then continued in an automatic mode. Full automatic mode was underway, with the technical support of CINECA. At the beginning of each cycle the atmospheric forcing data and the Po river discharge were downloaded from Arpae data repositories. Atmospheric data were then interpolated into the model grid.

Since the beginning of the operational activities, 2 day forecast runs for the physical and biogeochemical state of the northern Adriatic Sea were produced. A subset of the forecast outcomes was published daily on the "Marinomica" platform (https://marinomica. com/ (accessed on 30 June 2022)), including sea surface temperature, salinity, current velocity, dissolved oxygen, phosphate, chlorophyll, dissolved oxygen at the water–sediment interface and in the benthic interstitial water, and benthic filter feeder biomass.

### 2.5. The Validation Datasets

In order to achieve a preliminary assessment of the forecasting system performance, the results of the forecast/analysis runs were validated against remotely sensed satellite and in situ data collected by observing systems operating in the region (see below Sections 2.5.2 and 2.5.3). The distribution of fixed buoys and monitoring stations is shown in Figure 4.

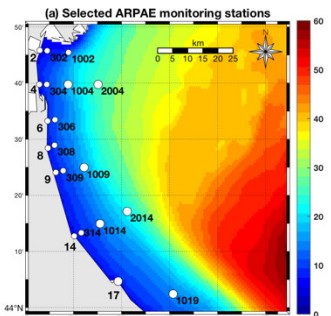
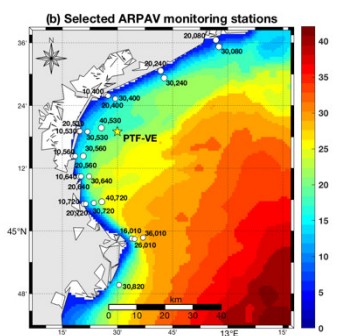
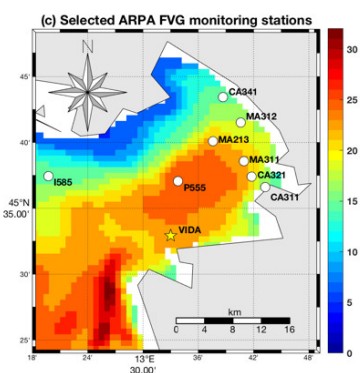

**Figure 4.** Structure of the Northern Adriatic coastal observing systems used to validate the forecasting system. (**a**) Arpae; (**b**) ARPAV; (**c**) ARPA FVG. The yellow stars in (**b**,**c**) indicate the location of the fixed buoys whose data were used for the validation procedure. See the text for details.

### 2.5.1. Remote Observations

Satellite data products obtained from the CMEMS marine service were the Level 4 Sea Surface Temperature (SST) and Chlorophyll (CHLA) fields, identified as SST_MED_SST_L4_REP_OBSERVATIONS_010_021 [47] and OCEANCOLOUR_MED_CHL_L4_REP_OBSERVATIONS_009_078 [48], respectively.

The SST product consists of daily (nighttime), optimally interpolated, satellite-based estimates of the foundation SST (namely, the temperature-free, or nearly-free, of any diurnal cycle) at 0.05° of horizontal resolution.

The CHLA product includes the daily interpolated chlorophyll field with no data voids starting from the multi-sensors MODIS-Aqua, NOAA-20-VIIRS, NPP-VIIRS, and Sentinel3A-OLCI, with a horizonal resolution of 1 km.

### 2.5.2. Observations from Fixed Stations

Continuous temperature and salinity observations are available from 2 buoys operating in the northern Adriatic Sea (yellow stars in Figure 4b,c): the Acqua Alta tower, operated by the CNR Marine Sciences Institute (ISMAR-CNR), located in the Gulf of Venice [49], and the VIDA buoy, operated by the Slovenian National Institute of Biology (NIB), located along the Slovenian coastal area in front of Piran [50]. The Acqua Alta tower observations are available at two different nominal depths (1.8 and 6 m), while data from the VIDA buoy are provided at the nominal depth of 2.5 m. The Acqua Alta data are available for the entire period of the operational activities, while the VIDA data are available only from January 2022 onwards.

The Acqua Alta tower's data acquisition averaging period is 10 min, while the VIDA buoy has an averaging period of 30 min. For the VIDA buoy, the provided data were already quality-checked, while the Acqua Alta tower data were manually quality-checked to remove spurious values. In both cases, data were averaged to daily resolution in order to be coherent with model output.

### 2.5.3. Observations from Coastal Observing System

In situ data over the period November 2021–April 2022 were collected in the framework of the observing systems operated by the regional environmental protection agencies, focusing on 3 specific coastal areas of the NAD. The structure of these observing systems is reported in Figure 4 and consists of the following:

- Arpae (Regional Agency for Prevention, Environment, and Energy of the Emilia–Romagna Region, Figure 4a) observing system, composed of 29 monitoring stations located south of the Po river delta (Emilia–Romagna coastal region) and sampled with a 15 day periodicity.
- ARPAV (Environmental Protection Agency of the Veneto Region observing system, Figure 4b), composed of 28 monitoring stations, located in front and north of the Po

River delta (along the Veneto coastal Region) and sampled during November 2021 and March 2022.

- ARPA FVG (Environmental Protection Agency of the Friuli Venezia Giulia Region, Figure 4c) observing system, composed of 21 monitoring stations, located within the Gulf of Trieste and sampled with a monthly periodicity.

These observing networks provide vertical profiles for temperature, salinity, chlorophyll, and dissolved oxygen, with the exception of chlorophyll data from Arpae, limited to the sea surface. Observations from Arpae also include surface-dissolved inorganic nutrient concentrations.

All the available surface samples within the model domain were considered for the validation procedure, while only the sampling stations having a nominal depth greater than 11 m (26 stations in total) were considered for the vertical profile comparison.

## 3. Results

Validation of model results against satellite-based observations was performed by interpolating the satellite data over the model grid, while validation against in situ sampling stations was performed by interpolating the model results to the observation locations using a nearest-neighbor algorithm.

### 3.1. Comparison with Remote Observations

Figure 5a,b respectively show the time series of the daily basin averaged values of SST and CHLA as obtained from the NAD-FC system and satellite-based observations. In order to compare the remotely sensed CHLA data with the model outcomes, the forecasted CHLA fields were averaged over the Photosynthetically Active Radiation e-folding depth. The comparison shows qualitative agreement, although the forecasted SST was about 1 °C lower than the satellite one. The average forecasted CHLA indicated an overestimation of about 1 mg Chl/m$^3$ in the first 3 months, while the difference decreased in the second part of the considered period.

Despite these differences, the Taylor diagram representation of the monthly basin averages for both analysis and forecast runs showed that SST has a very high correlation (above 0.90) and a normalized standard deviation between 1.1 and 1.4 (Figure 5c). Additionally, the monthly averaged CHLA has a high correlation (between 0.8 and 0.9) with the corresponding satellite data. However, the diagram illustrates how the forecasted CHLA has a much more variable degree of spatial variability, as indicated by the scattered values.

Since no data assimilation is currently performed for NAD-FC, the differences of the monthly forecast fields with respect to the analysis were minimal for both SST and CHLA. Therefore, no further quality assessment between forecast and analysis fields was carried out.

The differences between forecasted and satellite data can be interpreted in terms of spatial variability. Figure 6 shows the forecasted SST and CHLA fields obtained by averaging the 2 day lead forecasted fields over the period January–April 2022 (Figure 6a,d, respectively) along with the corresponding remotely sensed (Figure 6b,e, respectively) and bias field (Figure 6c,f, respectively). The qualitative agreement between the spatial variability of the forecasted (Figure 6a) and remotely sensed SST (Figure 6b) accounts for the high correlation shown by the Taylor diagram (see Figure 5c). The corresponding bias field denotes the overall tendency of the model to underestimate SST, in particular along the western and northern coastal regions, which are largely affected by the river runoff.

The forecasted surface CHLA (Figure 6d) indicated a weaker similarity with the remotely sensed field (Figure 6e), with marked positive biases occurring in the northern and western coastal areas, particularly south of the Po river delta. In the open sea area the overestimation was significantly lower.

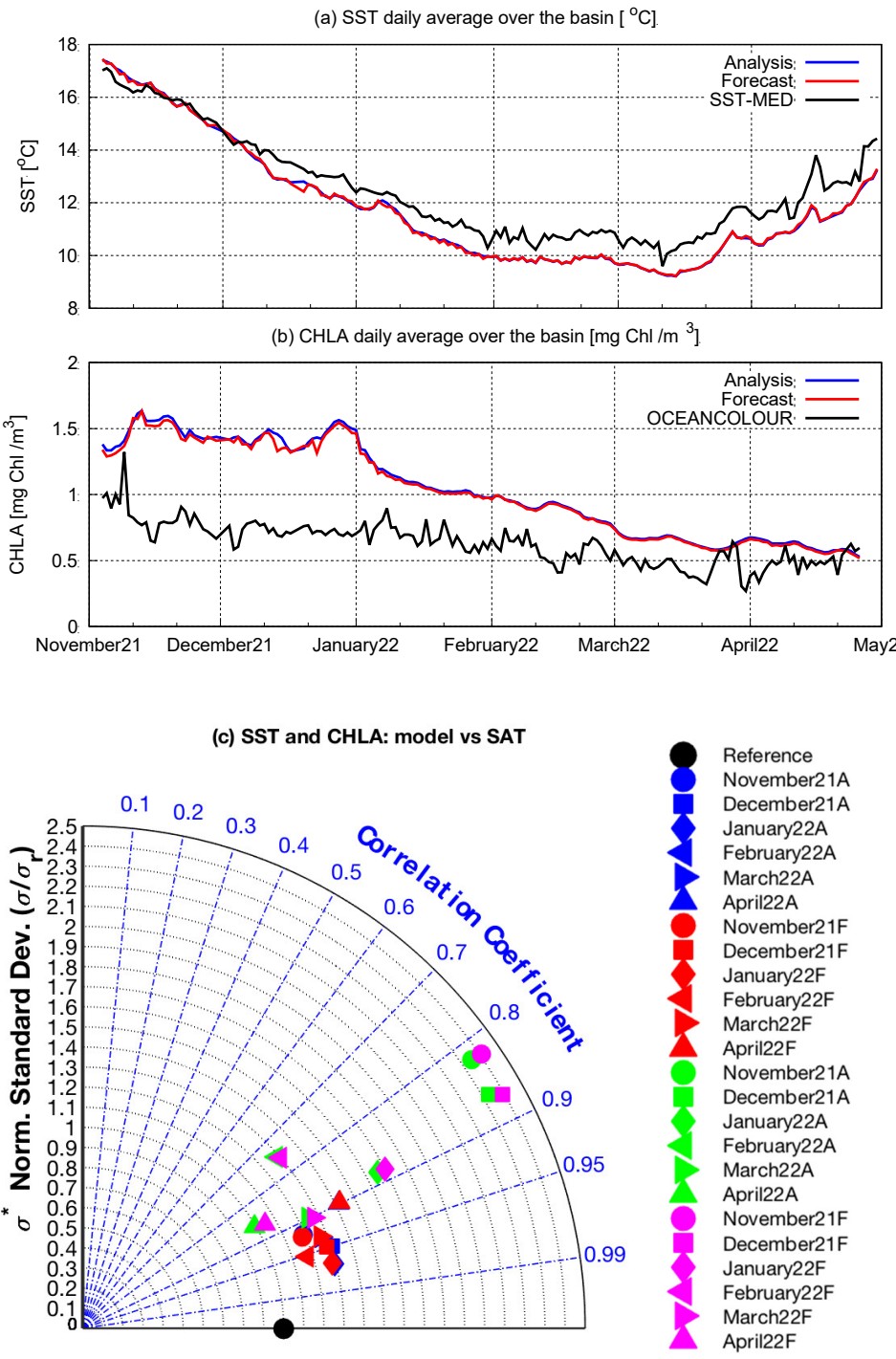

**Figure 5.** Time series of basin-averaged daily analysis (blue line) and forecasts (red line) of SST (**a**) and CHLA (**b**) and the corresponding satellite observations (black line). (**c**) Taylor diagram of monthly basin averaged SST (blue and red for analysis and forecast, respectively) and CHLA (green and magenta, for analysis and forecast, respectively) against the corresponding satellite observations.

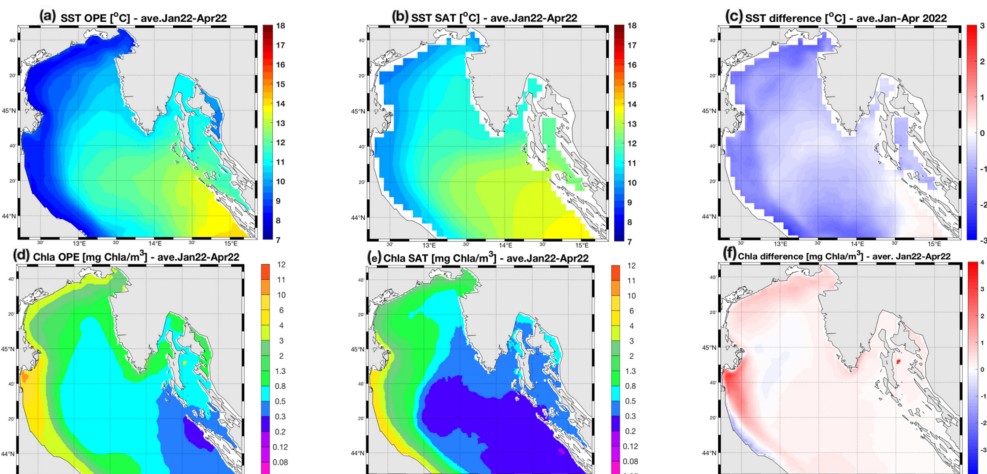

**Figure 6.** Maps of forecasted (OPE) and remotely sensed (SAT) SST and CHLA, averaged over the period January–April 2022. (**a**) Forecasted SST and (**b**) remotely sensed SST; (**c**) SST bias (OPE-SAT); (**d**,**e**) forecasted and remotely sensed CHLA respectively; (**f**) CHLA bias (OPE-SAT).

### 3.2. Comparison with Buoys Observations

The comparison of the model temperature daily time series with the corresponding VIDA and Acqua Alta observations is shown in the additional material, Figure S1a (VIDA) and Figure S1b,c (Acqua Alta). This gives a further confirmation of the already stated model tendency to underestimate the seawater temperature. The salinity time series are reported in Figure 7a (VIDA) and Figure 7b,c (Acqua Alta). The comparison of daily salinity values obtained from the NAD-FC with the measurements collected at the two buoys (Figure 7) indicated an overall tendency of the modeling system to overestimate the salinity value. Interestingly, the comparison also indicated a different model performance at the two sites. At the VIDA site, the salinity value was not affected by a marked time variability and the salinity bias remained relatively limited and almost systematic; at the Acqua Alta site, the time series indicated a larger time variability that the model was not properly capturing at both sampling depths.

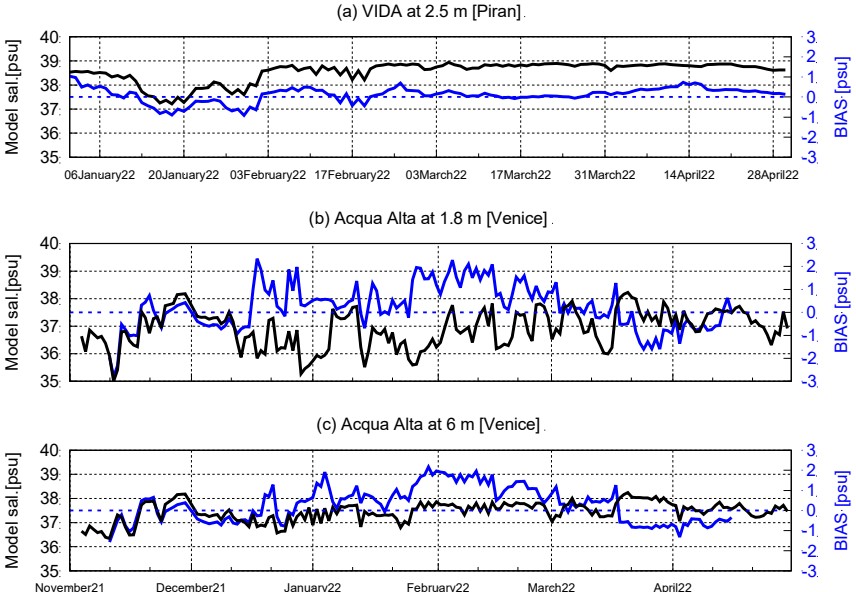

**Figure 7.** Comparison with salinity from buoys data. (**a**) Time series of forecasted salinity and bias with respect to the corresponding observations at the VIDA and (**b**,**c**) Acqua Alta locations. The black line indicates the forecasted values, while the blue line indicates the bias with respect to the buoy observations. The dashed blue line indicates the zero value of the bias axis.

### 3.3. Comparison with Data from Observing Systems

The available observation datasets of physical and biogeochemical state variables, operated by the observing systems described in Section 2.5.3, allowed for a thorough and challenging comparison that includes not only surface values but also vertical profiles.

The comparison with surface values by using the remotely sensed data, presented earlier, is here integrated by an assessment of the model performance with respect to the observed surface physical (temperature and salinity) and biogeochemical variables (chlorophyll, dissolved oxygen, and phosphate). Results are presented in the form of scatterplots in the model–observation space.

The temperature scatterplot of Figure 8a indicates a very good performance of the forecasting system in capturing the SST variability. Despite the slight underestimation tendency (y-axis intercept at about 1 °C), data from all the observing systems were close to having a linear relationship. This is more evident when considering the different position of the line fitting the available data with respect to the ideal fit, being the slope of the fitted line close to unity (1.02 °C).

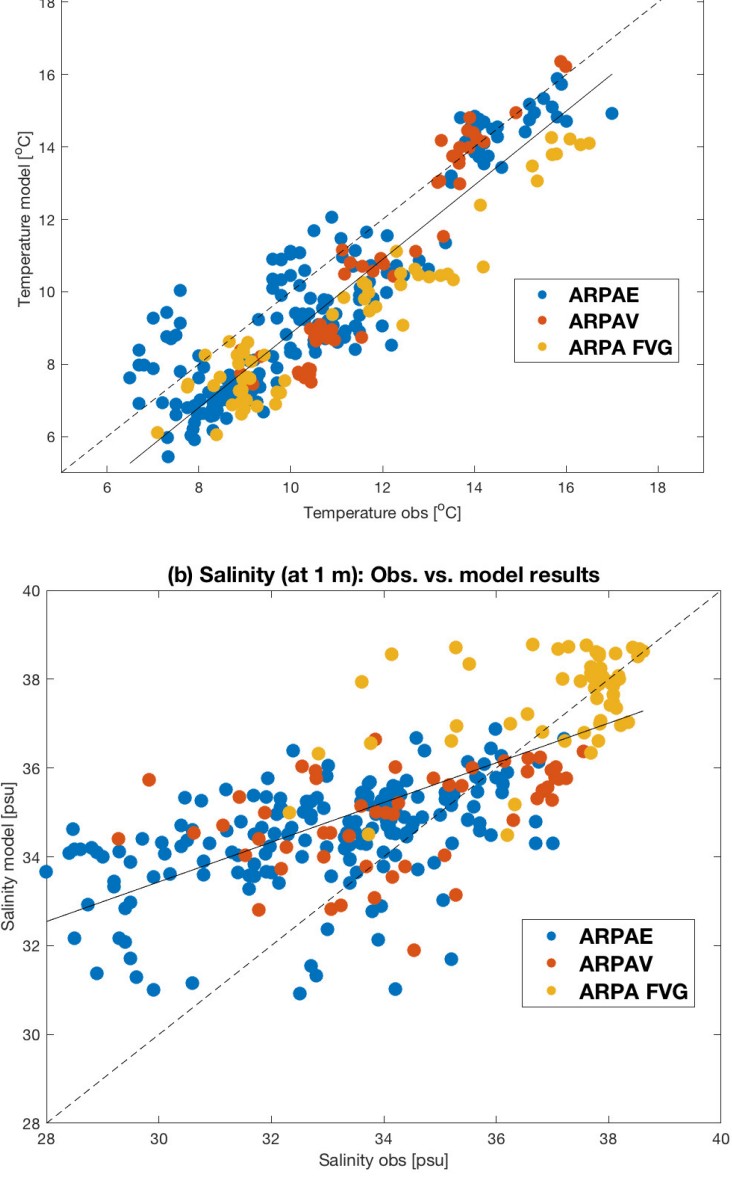

**Figure 8.** Scatterplot of forecasted surface physical variables versus observing system data. (**a**) Temperature; (**b**) salinity.

The surface salinity scatterplot of Figure 8b still indicates a roughly linear relationship between forecasted and observed data, but the fitted line diverges quite sensibly from the ideal fit. A marked model overestimation was quite evident for the coastal areas covered by the Arpae and ARPAV observing systems (i.e., the areas characterized by the mostly marked ROFI characteristics), while the comparison with ARPA FVG data showed a better fit.

Moving from the physical to the biogeochemical state variables, Figure 9 shows the scatterplot of surface CHLA and surface dissolved oxygen concentrations. The surface–chlorophyll comparison indicates a marked model overestimation (y-axis intercept at about 2.3 mg Chl/m$^3$), which was clearly determined by the large mismatch with the data of the Arpae observing system (blue dots in Figure 9a), mainly located south of the Po delta region. The wide scattering of the data within the model–observation space for this region led to an overall overestimation, as indicated by the blue line that fits the comparison with Arpae observations only. Conversely, the portion of the model domain covered by ARPAV and ARPA FVG (red and yellow points plus the black line) accounts for a model-to-observation relationship closer to the ideal fit.

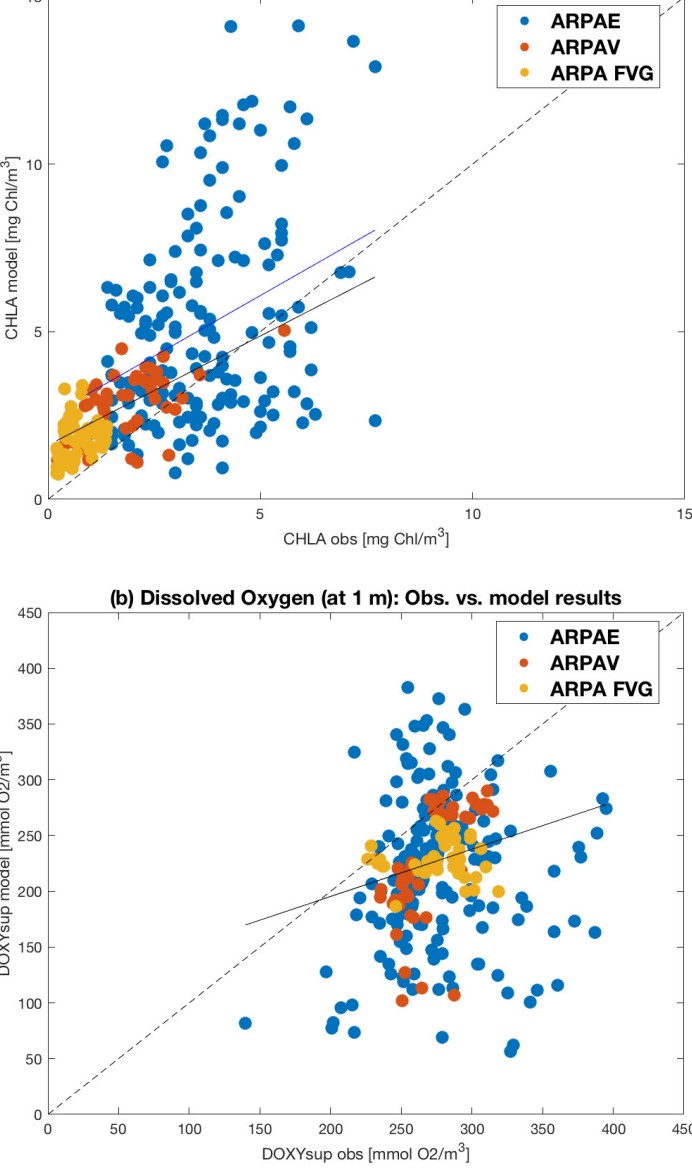

**Figure 9.** Scatterplot of forecasted surface biogeochemical variables versus observing system data. (**a**) Chlorophyll; (**b**) surface dissolved oxygen. Dashed and solid lines are as in Figure 8. The blue solid line represents data fitting for the Arpae data only.

The largely scattered points in the model–observation space arising from the comparison of the forecasted surface oxygen concentration and the corresponding Arpae observations provide a further indication that the Po river climatological parameterization of the nutrient load (see Section 2.3.1) is not satisfactory.

The analysis of the model performance dealing with the surface dissolved oxygen concentration opens the way to the analysis of the model performance with respect to subsurface properties. It is particularly relevant to assess the model behavior relative to a crucial state variable that defines the functioning of the Northern Adriatic coastal ecosystem, i.e., the dissolved oxygen concentration at the bottom of the water column. Together with the oxygen concentration in the sediment interstitial water, this is an important indicator of the occurrence of hypoxia/anoxia events, often affecting the coastal Adriatic Sea ecosystem.

Figure 10 shows the scatterplot of the forecasted and observed dissolved oxygen concentration at the water column lower layer. The plot indicates a linear relationship between observations and forecast, whereas observation data are generally underestimated. As for previous variables, the data within the area covered by the Arpae observing system showed a marked scattered distribution.

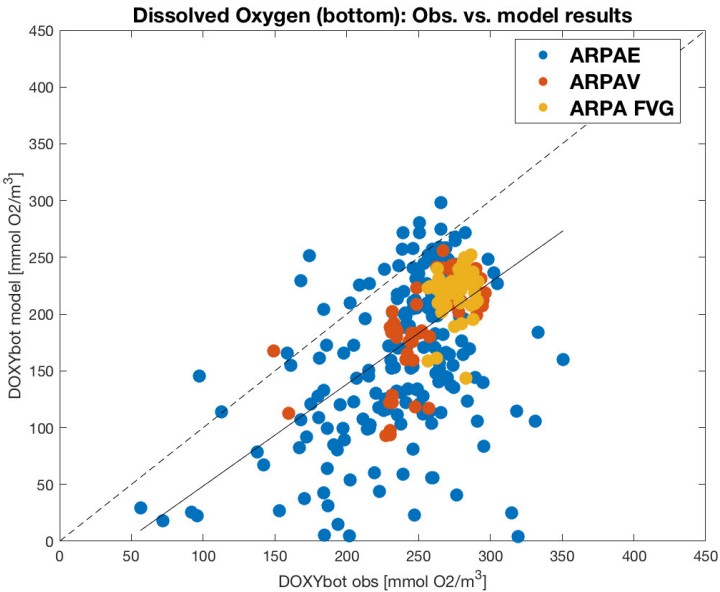

**Figure 10.** Scatterplot of forecasted dissolved oxygen at the water–sediment interface versus the observing system.

Such a deviation from linearity of the modeled chlorophyll and dissolved oxygen in the coastal region mostly affected by the Po river discharge is even more evident when considering the modeled surface phosphate concentration with respect to the in situ observations (Figure 11).

The discrepancy relative to the biogeochemical state variables (phosphate, chlorophyll, dissolved oxygen) in a region of the model domain dominated by the land-based fresh water and nutrient load fluxes points to a discrepancy with respect to real conditions in the definition of the lateral boundary conditions for the biogeochemical model.

The model performance in forecasting the observed vertical profiles relative to the Arpae and ARPA FVG systems is shown in Figures 12 and 13, respectively, by means of the model–observation bias and Pearson's correlation values computed for each profile ordered across time. In the Arpae region, both temperature (Figure 12a) and salinity (Figure 12c) were characterized by negative and positive biases, but the relationship between forecasted and observed profiles led to a good correlation (above 0.8) at most of the stations.

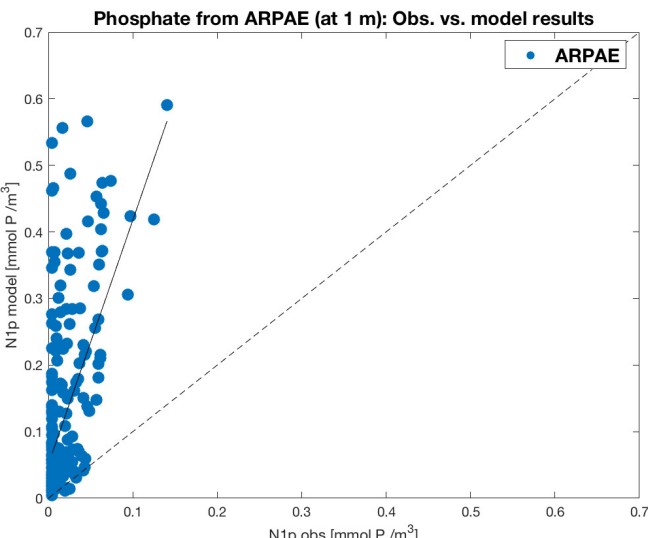

**Figure 11.** Scatterplot of forecasted surface phosphate versus observing system data.

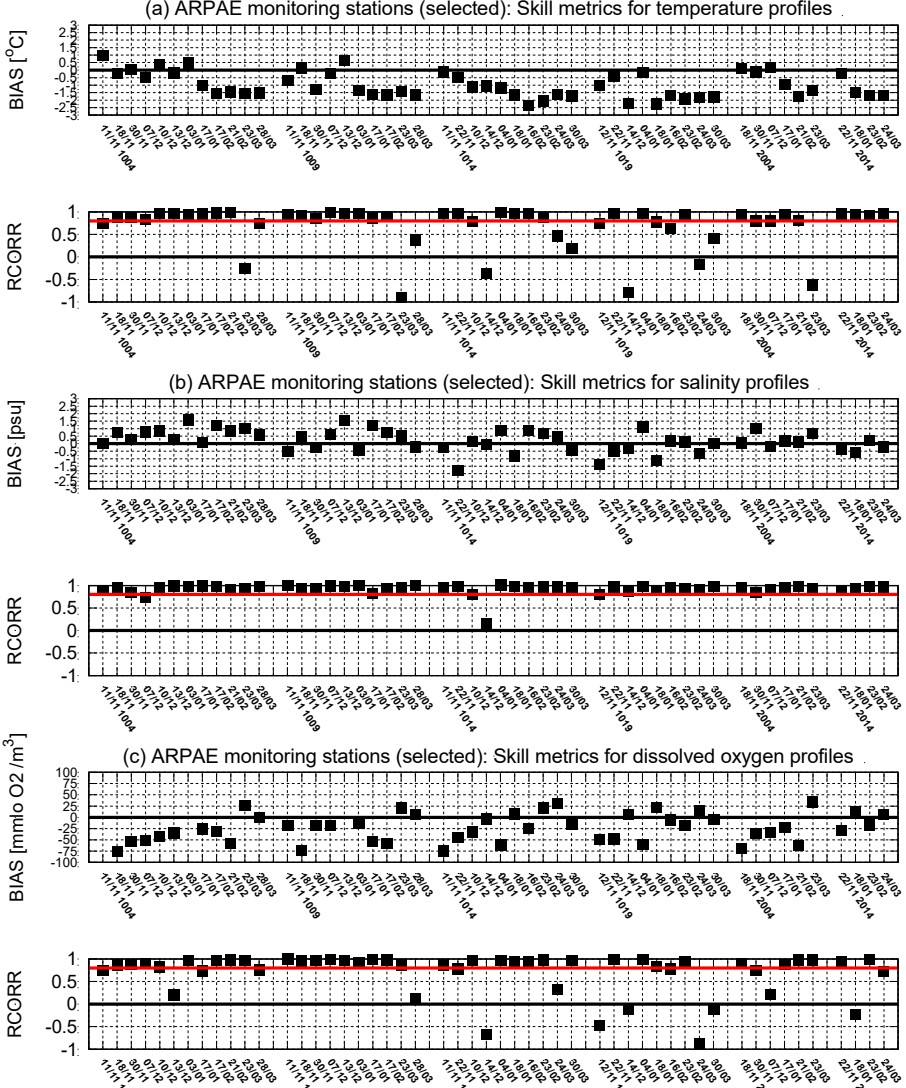

**Figure 12.** Skill metrics (bias and correlation) of the modeled vertical profiles (Arpae stations). (**a**) Temperature; (**b**) salinity; (**c**) dissolved oxygen. Black lines indicate the "zero" bias or correlation. The red line in the correlation plot indicates a correlation coefficient of 0.8.

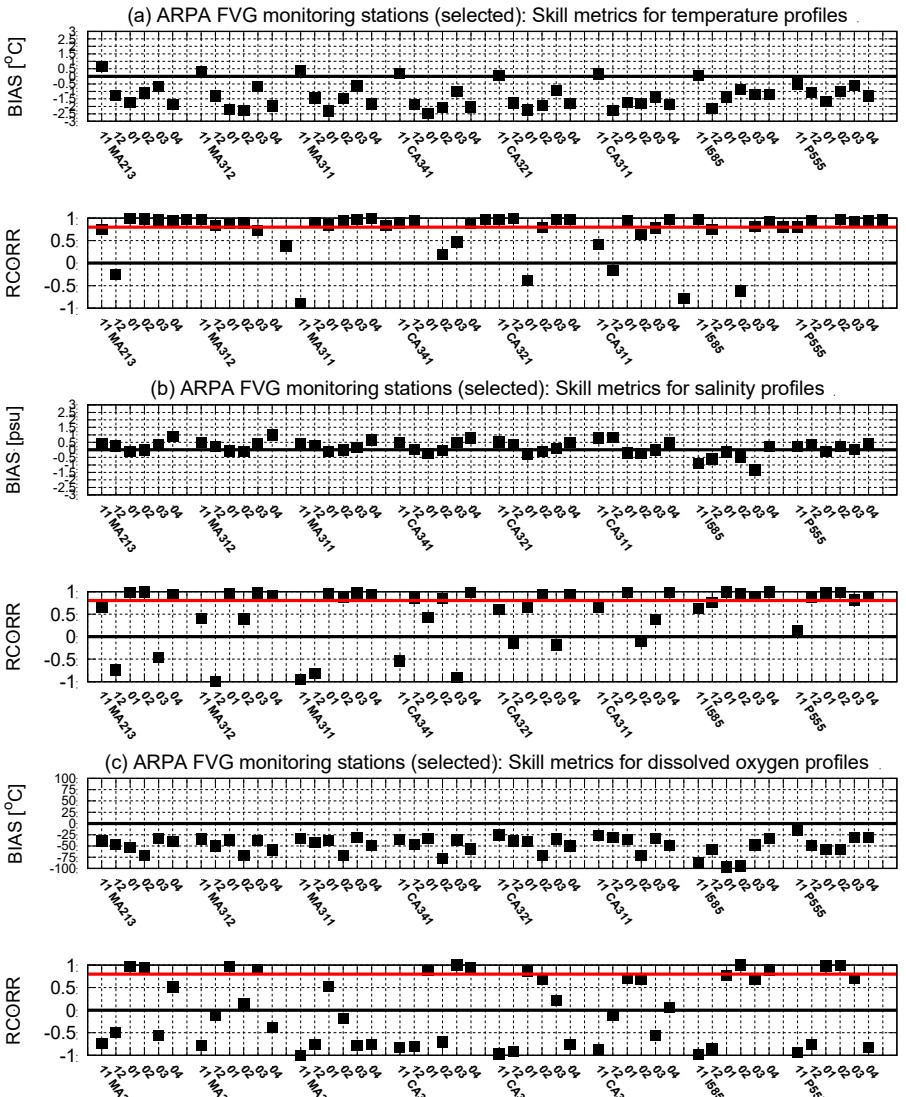

**Figure 13.** Skill metrics (bias and correlation) of the modeled vertical profiles in the Gulf of Trieste (ARPA FVG stations). (**a**) Temperature; (**b**) salinity; (**c**) dissolved oxygen. Black and red lines are as in Figure 12.

Figure 13 shows the same metrics for the area covered by the ARPA FVG observing system (see Figure 4b). The seawater temperature (Figure 13a) and salinity (Figure 13b) showed a marked bias tendency, namely negative for temperature and positive for salinity, and the temperature bias was rather high (up to −2.5 °C), while the salinity one was more limited. The correlation was still satisfactory for most of the stations, although certain profiles were clearly anticorrelated.

The dissolved oxygen (Figures 12c and 13c) bias was always negative, with most values between −20 and −40 mmol $O_2/m^3$. Differently from the hydrological state variable, the profile correlation was consistently and systematically high for the Arpae region, while it was quite variable (with positive and negative values) in the ARPA FVG region.

### 3.4. The Benthic Variables

The inclusion of the benthic dynamics in the 3D biogeochemical modeling of a shallow coastal basin, such as the northern Adriatic Sea, represents an innovative application in the field of coastal and shelf biogeochemistry. In fact, benthic biogeochemical processes can significantly constrain the coastal pelagic environmental dynamics through the inclusion of the biogeochemical cycling of the water sediment exchanges operated by truly chemical or

biologically mediated processes [33]. However, such a modeling effort is particularly challenging due to the lack of information on the biogeochemical benthic dynamics. Particularly important in the northern Adriatic Sea is the role and the impact of the benthic processes in defining the oxygenation levels at the lower end of the water column and in the sediment interstitial waters [51]. In fact, it is well known that sectors of the northern Adriatic coastline (particularly on the western side of the basin) can be subject to hypoxic-to-anoxic events [12,52].

In order to provide a preliminary overview of the behavior of the forecasting model with respect to the forecasted benthic state variables, Figure 14 reports the maps of the distribution of dissolved oxygen concentration in the sediment interstitial water (Figure 14a,b) and the distribution of the total biomass of the benthic fauna (Figure 14c,d). Fields are referred to periods close to the beginning and the end of the operational activities considered in this paper.

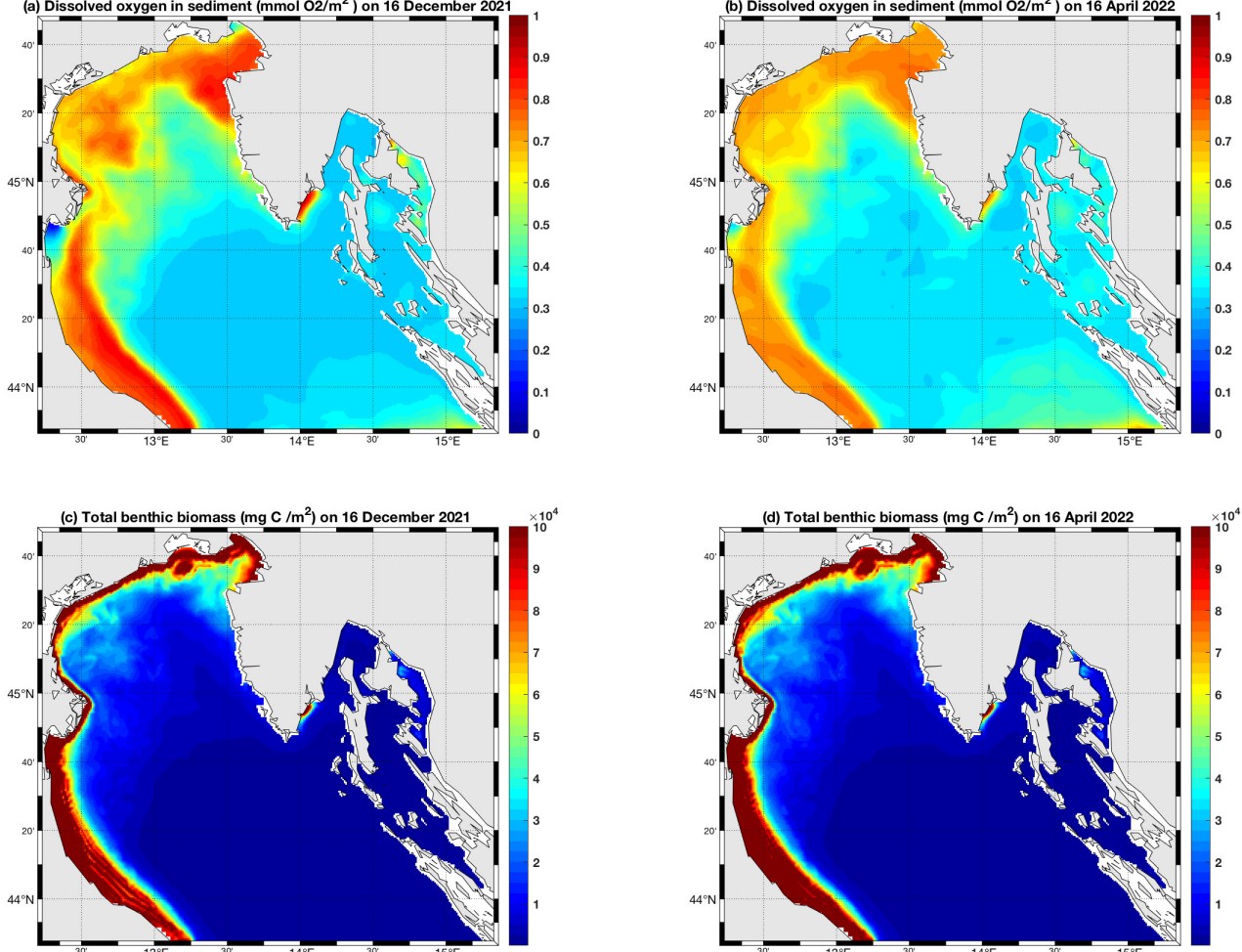

**Figure 14.** Maps of modeled benthic variables. (**a**,**b**) Dissolved oxygen in the sediment interstitial water on 16 December 2021 and 16 April 2022. (**c**,**d**) Total concentration of benthic biomass (mg C/m$^2$).

The distribution of the interstitial water dissolved oxygen concentration indicated that the oxygenation condition of the sediments was directly related to the bottom depth, namely the shallower the water column, the higher the sediment ventilation. During the operational period considered here (spanning from winter 2021 to early spring 2022) no particularly low oxygen concentrations were experienced over the whole model domain, mainly because of the well-mixed water column conditions facilitating sediment ventilation. However, the shallow region along the western Adriatic coast, immediately south of the

Po delta, was characterized by low benthic oxygen concentration values. This apparently indicates that the general configuration is potentially able to capture the variability of the benthic domain environmental conditions.

Assessing the model behavior with respect to the biomass of the benthic fauna is an even more challenging task. This issue is not only due to the lack of information on the consistency and on the spatial and temporal variability of the benthic fauna functional groups [53], but also arises from the fact that they are at the same time contributors and impacted actors of the benthic pelagic coupling. See, for instance, the role played in benthic–pelagic coupling by the filter feeders functional group, as described and discussed by Gili and Coma [54] and subsequently demonstrated with a numerical model by Mussap and Zavatarelli [33].

The distribution of the total benthic biomass at the beginning and at the end of the considered operational period (Figure 14c,d) indicated that the general benthic population (defined as the sum of the biomass of the five functional groups listed in Figure 2) achieved a certain stable condition.

## 4. Discussion and Conclusions

The assessment relative to the first 6 months of operational activity of the NAD-FC indicates points of both strength and weakness of the current configuration of the system.

The temperature fields from forecast/analysis runs (Figure 6c) were generally well captured at the surface and throughout the vertical profiles, despite specific but systematic underestimation (see Figure 8a). The temperature field crucially depends on the heat exchanges at the ocean–atmosphere interface, so the good performance of the system indicates a good quality of the atmospheric analysis and forecast forcing of the modeling system. However, the underestimation probably requires an accurate evaluation of the bulk formulas used to compute the heat fluxes.

The salinity fields from the forecast runs can be compared only with in situ observations from local observing systems. The comparison indicates a general overestimation of the values at the surface (Figure 8b), which extends in depth, given the weaker (with respect to temperature), but still satisfactory, system performance in correlating with the observed vertical profiles (Figures 12b and 13b). As stated in the introduction, the northern Adriatic Sea is a sub-basin with ROFI characteristics, whose representation in the system forcing is still lacking realism. In fact, the configuration of the land-based freshwater forcing structure defines the Po river runoff at daily resolution, while the discharge from the other minor rivers is defined at the monthly climatological level. This is due to the difficulty to trace runoff observations timely available for operational use in the forecasting system forcing. Additionally, the a priori definition of the river water salinity at 15 psu might require a redefinition to embed at least spatially varying values.

The performance of the biogeochemical component of the forecasting system also appears to be impacted by the use of climatological values for the river-borne nutrient load. The availability of nutrient data from the Arpae observing system (the region most directly affected by the Po river discharge and nutrient load, see Figure 4a) allowed the evaluation of the concentration level of the phosphate field (Figure 11). This comparison revealed a marked model overestimation, which inevitably feedbacks on the primary producer biomass (expressed as chlorophyll concentration), which is also overestimated along the northern and western coastal areas (Figure 6f). The spatial distribution of the bias value, as well as the scatterplot of Figure 9a, indicate that the model deviation from observations was stronger in the areas directly affected by the river runoff and nutrient load. Despite the general overestimation discussed above, the forecasted surface chlorophyll concentration distribution at the basin scale was still in very good agreement with the satellite-based observations (Figure 5).

It has been previously mentioned that coastal regions of the northern Adriatic Sea might experience events of bottom hypoxia. Therefore, the model's ability to capture the variability in space and time of the oxygen state variable is a crucial feature for

its operational activities. The model's performance is not entirely satisfactory yet (see Figures 9b and 10), as the match with the observations is clearly affected by an evident data scattering. This scattering was particularly intense for the coastal area downstream the Po river delta, while the comparison with the data from the other two observing systems was slightly more satisfactory. This implies a problem in the forcing structure for the biogeochemical component of the system.

It can be concluded that the current state of the NAD-FC appears promising despite the shown weaknesses. The physical component allows reproducing, in a very correct way, the temperature field, thanks also to the quality of the atmospheric data used to force the model. The comparison with salinity is, however, affected by the approximate definition of the land-based fresh water and nutrient-load forcing. This appears to be a very general problem, as the model's weaknesses in replicating the biogeochemical state variables appear linked to uncertainties of the forcing.

From an operational modeling perspective, also taking into account that no data assimilation is performed, the NAD-FC current configuration delivers a satisfactory performance. Obviously the introduction of data assimilation of physical and biogeochemical variables will, hopefully, further improve the forecasting skill.

Finally, the inclusion of the benthic model component is a specific novelty in the field. The limited period of operational activity along with the lack of coherent observations do not allow a quantitative assessment of the performance. However, as a very preliminary and limited conclusion, it can be stated that the benthic component of the model is stable and is producing reasonable results.

**Supplementary Materials:** The following material is available online at https://www.mdpi.com/article/10.3390/w14172729/s1, Figure S1: model-fixed buoy temperature comparison. (a) VIDA buoy, (b) Acqua alta (1.8 m depth), (c) Acqua Alta (6 m depth). Black line model temperature, Blue line bias.

**Author Contributions:** Conceptualization, I.S. and M.Z.; methodology, I.S., M.Z. and T.L.; software, I.S., P.L., M.Z. and T.L.; validation, I.S., M.Z. and T.L.; resources, P.L. and A.V.; writing—original draft preparation, I.S. and M.Z.; writing—review and editing, I.S., M.Z. and T.L. All authors have read and agreed to the published version of the manuscript.

**Funding:** This research was funded by the European Union's Horizon 2020 research and innovation programme "ODYSSEA" (Operating a network of integrated observatory systems in the Mediterranean Sea) under grant agreement No 727277.

**Data Availability Statement:** Results from the operational forecasting system can be viewed at the site www.marinomica.com (accessed on 30 June 2022).

**Acknowledgments:** The authors wish to thank the Environmental Protection Agencies of Emilia-Romagna Region, Veneto Region and Friuli Venezia Giulia Region for providing data from the institutional monitoring programs. We also thank CNR-ISMAR of Venice and NIB Institute of Piran for data from the Acqua Alta tower and VIDA Buoy, respectively. This work is dedicated to the memory of our colleague Gelsomina Mattia.

**Conflicts of Interest:** The authors declare no conflict of interest. The funders had no role in the design of the study; in the collection, analyses, or interpretation of data; in the writing of the manuscript, or in the decision to publish the results.

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
