# Peer review of "The Northern Adriatic Forecasting System for Circulation and Biogeochemistry: Implementation and Preliminary Results"

_water, doi:10.3390/w14172729_

Round 1

Reviewer 1 Report

The overall impression is that the Manuscript presents an interesting idea with a lot of work put into the presented research. The issue is that the presentation of this research isn’t as good as it should be. In addition to the comments listed below,  I would suggest major editing of the paper as it quite confusing. It should be clearly stated what and where was measured, including the frequency and measuring methodology. Also, what model was used to model various segments of the data, with adequate description of what the implemented model did within this study. Furthermore, how were the models combined, what was the output from the models, and which measurements were used for validation with which of the measurements. These presentations should also include exact comparison of the numerical values wherever possible. Although all of this is done in some manner, most of the time the Authors simply direct the readers towards some literature instead of providing the needed information. In this way the paper seems more as a literature review type, instead of a research article presenting the implemented study and results.

Exact comments are listed below:

Referencing is incorrect and should be changed in accordance to the journal’s guidelines throughout the Manuscript.

Another issue noted throughout the paper is that often notations are used without being previously properly labelled. I would ask the Authors to make sure all of the used abbreviations and notations are properly described.

Figure 1 should be complemented with the exact names of all the rivers that are labelled as inflow. There is no point in marking them on the map if they are not all named so the reader can identify them.

Section 2. should be significantly complemented to include a more detailed explanation regarding the employed models, and equations they use. After reading the overall paper it seems interesting but hard to follow as it lacks exact information regarding the implemented research methodology. This part of the work should include more information that would allow the reader to understand what exactly the Authors did, instead of simply listing references of existing models.

Figure 2 should be labelled a) and b), as it is referenced in such a way but the images are not labelled.

Lines 121 to 128 indicate the Authors made some changes to the implemented model. Please include in detail what changes were made and ad appropriate equations not just explanations.

Section 2.3.should also be presented in more detail. The dimensions of the area should be given exactly (e.g. the area of the considered domain). The description of the are needs to be presented in more detail. It isn’t clear what sentence “The model internal time step is 120 seconds.” (Line 158) exactly means, additional explanation is needed.  

Line 172, sentence,  “The main River discharging into...”, please explain what is meant by the main river.

Figure 3 is blurred, and should be corrected.

Line 223 “... by different observing systems”, please explain these in more detail.

Figures 4 should be synchronised. Furthermore, Fig. B needs additional alterations as the numbers aren’t all visible.

Line 235, sentence “... consists of daily (night time)...” please explain what this means.

Figures 6 should also be amended, font size is too small.

Line 331, what are Figures S1a and S1b?

Line 383, delete one of the two surface surface words.

Line 525 , sentence “Therefore? the model ability...“ should be corrected to “Therefore? the model’s ability...“

Line 531, sentence “This points once more to...”,  should be amended “ This implies to a problem...”

Line 545 contains a link that leads to an error

Reviewer 2 Report

Please fine the attached review.

Round 2

Reviewer 1 Report

The revised Manuscript is improved as required and I would suggest publishing it. I would like to thank the Authors for their cooperation and fast reaction.

Reviewer 2 Report

The authors have effectively responded to my previous comments and the manuscript has been improved compared to the previous version. I have no further comments.